# Is It All about the Price? An Analysis of the Purchase Intention for Organic Food in a Discount Setting by Means of Structural Equation Modeling

**DOI:** 10.3390/foods9040458

**Published:** 2020-04-08

**Authors:** Felix Katt, Oliver Meixner

**Affiliations:** Institute of Marketing & Innovation, Department of Economics and Social Sciences, University of Natural Resources and Life Sciences, A-1180 Vienna, Austria; felix.m.katt@gmail.com

**Keywords:** organic food, discount supermarket, purchase intention, structural equation model (SEM), grocery retailing

## Abstract

In recent years, discount grocery retailers have expanded their global reach and added to their traditional no-frills offerings to also cater to hedonic consumer needs. In addition to a larger product assortment and a more pleasant shopping experience, they now sell organic food, which commands a price premium compared to non-organic alternatives. To understand organic food in a discount setting, this study sets out to examine the factors that influence discount grocery shoppers’ purchase intention for organic food. To study this relationship, this paper tests several factors in a structural equation model, finding a positive relationship between hedonic shopping values, environmental concern, as well as health consciousness and the purchase intention for organic food. In our model, based on a US consumer survey (*n* = 394), price consciousness exhibited a direct and negative relationship with the purchase intention for organic food. Furthermore, this study found that that the impact of environmental concern, health consciousness, and hedonic shopping value is greater on the purchase intention of organic food than that of price consciousness, even in this discount setting. This study concludes by discussing these results and attempting to outline potential areas for future research, as well as managerial implications.

## 1. Introduction

The success story of discount grocery retailing has been widely studied in the last two decades: from studies on their business model and internationalization strategies [1,2] to consumer price attribution [3], loyalty [4], and shopping value [5]. Historically known for low prices, traditionally undercutting other supermarkets and hypermarkets by 15 to 30 percent [1], limited product assortments, and efficient operations [6,7,8], the industry is now undergoing substantial changes. Over recent decades, the two leading companies in this industry, Aldi and Lidl, have shaped and cultivated the image of the “hard discounter” with little convenience, products piled in boxes, low price, private label products, and limited investment in technology [9]. With these characteristics, the concept has proven widely successful. From conquering Europe in the 1990s [1,6] to the current expansion overseas [10], both Aldi and Lidl have written a global success story. Through their business models, they have been able to make swift gains of market share in mature markets dominated by supermarkets with a strong brand and service orientation, for example, the United States [1]. Both companies have managed to appeal to what is now a very broad range of consumers, from the initial lower-income bargain hunters to today’s “hybrid consumers” [11], with higher incomes and selective spending preferences. Nevertheless, in their European core markets, Aldi’s and Lidl’s success has begun to slow down considerably. Therefore, discount grocery retailers have adjusted their business model in recent years—notably, by offering a broader and deeper product assortment, by introducing new (more premium) private label (PL) tiers, and by a steady increase in national brands (NBs) available to shoppers [2,12]. In addition to changes in their product assortment, many discount grocery retailers have moved toward a greener image, from investments in sustainability [13] to offering an increased number of organic food product choices [14,15].

It is thus critical for practitioners and scholars alike to understand what this organic food offering means for discount grocery retailers. The question is whether they can appeal to their core customers with this organic offering and whether they can win over environmentally-minded consumers preferring organic food, who might otherwise not frequent their stores. As consumer demand for organic food has increased in the past two decades, researchers have extensively studied the purchase behavior associated with organic food (see [16] for a systematic review). This behavior, however, has very rarely been studied in a discount retailing context, but rather, from a price-sensitivity [17,18] or frugality [19] perspective. In a study that examined organic food in a discount retail setting, Gottschalk and Leistner [20] found that discount grocery shoppers tend to be more price-sensitive when buying organic food, whereas repeat buyers are more strongly influenced by other product characteristics. They also found that the availability of organic food generally triggers a purchase, hinting at “supply [creating] its own demand” [20]. To add to these findings, this study aims to evaluate the drivers that influence discount grocery shoppers’ purchase intention for organic food, building a framework and testing it using a structural equation modeling (SEM) approach. Up until now, no comparable studies focusing on purchasing drivers for organic food within a discount grocery setting and using the SEM approach seem to have been published. Therefore, we aim to contribute to the literature by focusing on purchasing drivers for organic food within a discount grocery setting and using an SEM approach to help better understand customers’ purchase intention for organic food in discount grocery stores.

This study is structured into four distinct sections. It commences by outlining the conceptual model developed for this study, as well as the hypotheses to be tested. Next, the methodology employed is detailed, and the experimental design discussed. Thereafter, the results are presented, and the study is concluded with a discussion of the said results, potential future research avenues, managerial implications, and limitations.

## 2. Conceptual Model

There are numerous context-dependent factors that can influence consumers’ purchase intention and behavior in a shopping situation. To understand how the discount grocery retail setting might affect consumers’ purchase intention for organic food items, we draw on commonly cited drivers of purchase behavior traditionally associated with discount grocery retail (price consciousness), common drivers of organic food purchase intention (environmental concern and health consciousness), and an emerging factor in discount grocery retailing—hedonic shopping value. These form the basis of our conceptual model. Such an attitudinal approach to determining the drivers for purchase intention is grounded in the theory of reasoned action (TRA) developed by Fishbein and Ajzen [21], as this theory can be effective in explaining psychological and cognitive antecedents to decision-making [22] as well as attitudes serving as an important predictor of behavioral intention [23]; it is therefore frequently used in the organic food context [24,25]. These four individual drivers, as well as their hypothesized relationship with the purchase intention for organic food for this study’s model, are discussed in detail below. Figure 1 summarizes the conceptual framework for this study.

### 2.1. Hedonic Shopping Value

Discount grocery retailers, traditionally known for low prices and a no-frills service offering, have moved from only offering utilitarian shopping value to its consumers to also catering to hedonic needs [5] by offering a more stimulating in-store atmosphere and a broader product assortment. Nevertheless, the question of providing utility and value still forms the traditional basis of discount grocery retail. Utilitarian shopping value can be described as a task-oriented way of shopping, focusing on efficient outcomes without many emotions involved [26]. In the context of discount grocery retailing, this means giving consumers the impression of buying good value at low prices in a shopping environment that is not overly complicated (e.g., by providing limited in-store stimuli). In contrast to this, hedonic shopping value results from fun, enjoyment, and entertainment—that is, “hedonically rewarding shopping experiences are not akin to a negative sense of work” [26]. As organic food is often a more premium (i.e., more expensive) alternative to non-organic options, shoppers may not directly derive value in a cost or efficiency sense. However, with regard to organic food, consumer value perception may be based on a variety of factors other than price, i.e., they may derive hedonic shopping value. Generally speaking, discount grocery shoppers perceive the products they are able to buy to be of good value [27], and in this discount context, the said value perception generally holds a positive effect on purchase intention [10]. Therefore, we aim to understand what effect hedonic shopping value might have on organic food purchase intention in a discount grocery setting as a premium food category like organic could be considered a departure from the low-cost, no-frills commercial strategy. We propose that

**H_1_.***Hedonic shopping value has a direct and positive effect on the purchase intention for organic food in a discount grocery setting*.

### 2.2. Price Consciousness and Purchase Intention for Organic Food

Ever since their inception, discount grocery retailers such as Aldi and Lidl have attracted customers with low prices; their main customer base was, and probably still is, price-conscious consumers [6]. While, in general, grocery consumers tend to base their purchase decisions not only on price, but on a variety of attributes, such as quality [28], price consciousness can play a substantial role in purchase decisions [29], especially for the majority of shoppers who frequent discount grocery retailers [30]. As organic food items are generally more expensive than their non-organic alternatives [31], a high price can not only lead to a decreased purchase intention for organic food [32] but can also potentially act as a barrier to consumption all together [33]. Given the price sensitivity of discount grocery shoppers, we, therefore, hypothesize the following relationship:

**H_2_.***Price consciousness has a direct and negative effect on the purchase intention for organic food in a discount grocery setting*.

### 2.3. Environmental Concern and Purchase Intention for Organic Food

Environmental concern, defined by Schultz and colleagues [34] as a concern about environmental challenges caused by human behavior, is an oft-cited driver for organic food purchase behavior with a positive influence on purchase intention [35], actual purchase behavior [36], and the willingness to pay a price premium [37,38]. This may be explained by the finding that organic food consumers also engage in eco-friendly behavior, such as food waste reduction [39] and recycling [40]. As environmental concern evolved to be a more mainstream topic of public discourse [41], discount grocery retailers started to shift toward a greener image. Given the effects of consumer environmental concerns, we would like to understand its impact on the purchase intention for organic food in a discount grocery retailing setting. Thus, we propose that

**H_3_.***Environmental concern has a direct and positive effect on the purchase intention for organic food, even in a discount grocery setting*.

### 2.4. Health Consciousness and Purchase Intention for Organic Food

Health consciousness can be an important factor in consumer food purchase decisions [42], especially with regard to organic food items. Several studies have not only found health consciousness to be a driver for organic food purchases [43,44], but also a strong influence on consumers’ willingness to pay an organic premium [44,45], perhaps explained by consumers perceiving organic food alternatives to deliver greater health benefits [46]. Analogous to environmental concern, we aim to understand the role of health consciousness in the context of discount shopping behavior for organic food. Therefore, we hypothesize the following relationship:

**H_4_.***Health consciousness has a direct and positive effect on the purchase intention for organic food, even in a discount grocery setting*.

### 2.5. Other Interactions

In addition to these main effects, we aim to understand the interaction between the proposed main drivers in a grocery discount setting, i.e., hedonic shopping value and price consciousness, and the interaction between the established main drivers for the purchase intention for organic food. Given that discount grocery retailers main customer base was and probably still is price-conscious consumers [6], we suspect that this diminishes the hedonic shopping value that may be derived for consumers in such a discount setting even in light of the aforementioned move of discount grocery retailers to also cater to hedonic needs [5]. Furthermore, we expect that the other proposed drivers for discount grocery shoppers’ purchase intention for organic food exhibit a positive relationship given their importance in organic food consumer behavior in general [16], especially given the findings of Tsakiridou et al. [47], as well as De Magistris and Gracia [48], who uncovered a link between attitudes towards one’s health and the environment in organic food consumers. Nevertheless, this might be contrasted by Gschwandtner [31], who found that only health but not environmental friendliness leads to a higher organic food willingness to pay.

We thus propose that

**H_5_.***Hedonic shopping value is negatively correlated with price consciousness in a discount grocery setting*.

**H_6_.***Environmental concern is positively correlated with health consciousness in a discount grocery setting*.

Altogether, the conceptual framework of this study leads to Figure 1, visualizing the hypothesized relationships between the independent and dependent variables.

## 3. Methodology

### 3.1. Data Collection

For this study, we employed a questionnaire survey to collect data to analyze the developed conceptual framework. The research setting was the United States, a country with a diverse grocery retailer landscape. Data was collected through Amazon Mechanical Turk (MTurk), a frequently used crowdsourcing platform for human tasks such as surveys. MTurk allows researchers to anonymously recruit study participants based on pre-selected criteria such as country of residence and is frequently used in consumer survey research in the organic food sector [49]. To ensure reliable and valid results, we followed previous researchers’ guidelines [50,51] for recruiting respondents through MTurk and employed measures such as restricting survey-takers to respondents with high MTurk approval ratings. The shopping behavior at discount supermarkets (in this study, i.e., Aldi and Lidl) was self-reported at the end of the survey, leading to an eligible initial sample of 411 participants. This self-reported shopping behavior was elicited by asking respondents to select their most frequented grocery retailers to ensure that discount grocery shoppers were adequately captured. Of this initial sample of 411, 17 respondents were excluded due to failing an attention check, for straight-line answer patterns, or for not completing the majority of the survey, resulting in a final sample of 394 (response rate = 95.9%). The sample is skewed toward the more educated share of the US population, but is in line with the median household income [52] and mean age (38.2 years) [52]. Table 1 provides an overview of the demographics of the sample.

### 3.2. Measurement Instruments and Analysis

To employ valid measurement instruments, the scales used in this study were adapted from previous studies. The scales used were measured on a 7-point Likert scale, with a score of one denoting “strongly disagree” and a score of seven denoting “strongly agree”. The statistical analysis was conducted using the software solutions SPSS (Statistical Package for Social Sciences, version 26) and AMOS (Analysis of Moment Structures, version 26, Mount Pleasant, SC, USA). First, an exploratory factor analysis (EFA) was conducted to test the factorial structure of the selected items from the questionnaire to develop reliable multi-item scales. The environmental concern (Cronbach’s α = 0.736) scale, adapted from Wei, Ang, and Jancenelle [49] and De Magistris and Gracia [48], was operationalized with four statements, as was price consciousness (Cronbach’s α = 0.801), the items of which were adapted from Gil and Soler [53]. Health consciousness (Cronbach’s α = 0.902), which was adapted from Michaelidou and Hassan [54], was operationalized with six statements, as was hedonic shopping value (Cronbach’s α = 0.955), using the items developed by Babin et al. [26]. Lastly, the purchase intention for organic food was adapted from Yadav and Pathak [55] and operationalized with five statements. 

The details of the measurement instruments, as well as their sources, are shown in Table 2. Following the EFA, we conducted a confirmatory factor analysis (CFA), validating the measurement model, as well as testing, fitting, and modifying the structural model. The developed hypotheses were tested by way of standardized regression coefficients (β) and *p*-values (*p*). The results are discussed in detail in the next section.

## 4. Results

Generally, the respondents answered highly for price consciousness (mean = 5.757, SD = 0.912) questions, as well as showing a positive attitude toward their health (mean = 5.581, SD = 0.955) and toward the environment (mean = 4.806, SD = 1.235). The purchase intention for organic food was also slightly positive (mean = 4.838, SD = 1.523). The participants’ responses for the hedonic shopping value factor (mean = 4.230, SD = 1.672) were close to neutral. Figure 2 visualizes the responses received, showing a homogenous response pattern for all constructs except for environmental concern, which may be explained by employing two reverse-coded items, which tend to act as cognitive “speed bumps” for respondents, behaving differently from the normal (positively) coded items [56]. The two reverse-coded items were used to reduce social desirability bias for the resulting environmental concern construct, which exhibits good validity (Cronbach’s α = 0.736).

### 4.1. Measurement Model

The CFA model was operationalized after deleting one environmental concern item (EC2, “I can think of many things I’d rather do than work toward improving the environment”), which improved the average variance extracted (AVE) for the environmental concern construct above the 0.500 threshold proposed by Hair, Black, Babin, and Anderson [57]. The resulting measurement model exhibited good fit indices: χ^2^ = 712.034, degrees of freedom (df) = 242, Tucker-Lewis Index (TLI) = 0.924, comparative fit index (CFI) = 0.933, root mean square error of approximation (RMSEA) = 0.070, and standardized root mean square residual (SRMR) = 0.054. The regression coefficients corresponding to all the measurement items were significant (*p* < 0.001). This is in line with the recommended cutoff values for SEM [57,58].

Next, reliability and validity were assessed. Regarding convergent validity, all factors exhibited good composite reliability (CR), with values ranging from 0.772 to 0.956, and higher than the AVE, which for all factors was above the 0.500 threshold. Regarding discriminant validity, all AVEs are greater than the maximum shared variance (MSV), and the square root of the AVEs is greater than the inter-construct correlations. Table 3 provides an overview of reliability and validity measures.

To account for method bias [56], we conducted an additional CFA for the constructs and an added common latent factor. The results show that the latent factor accounts for 24% of the total variance, below the typical method variance found by Williams, Cote, and Buckley [59]. This model also exhibited good fit indices (χ^2^ = 707.425 df = 241, TLI = 0.924, CFI = 0.934, RMSEA = 0.070, SRMR = 0.060).

### 4.2. Structural Model

The results of the structural model exhibited good fit indices (χ2 = 712.034, df = 242, TLI = 0.924, CFI = 0.933, RMSEA = 0.070, SRMR = 0.054). The first hypothesized relationship was the direct and positive relationship of hedonic shopping value with the purchase intention for organic food. H_1_ of our model is supported (β = 0.226, *p* < 0.001). In addition to this relationship, the hypothesized (H_3_) direct and positive relationship of environmental concern with the purchase intention for organic food was supported (β = 0.128, *p* = 0.016). Similarly, we found a direct and positive relationship between health consciousness and the purchase intention for organic food (β = 0.459, *p* < 0.001), supporting H_4_. Next, we examined the relationship between price consciousness and the purchase intention for organic food, finding a negative and significant relationship (β = −0.130, *p* = 0.022), thus supporting H_2_. Lastly, we examined the hypothesized interactions between the exogenous variables. While we did not find empirical support for H_5_, the hypothesized negative correlation between hedonic shopping value and price consciousness (*r* = 0.014, *p* = 0.809), we found a significant positive correlation between environmental concern and health consciousness (*r* = 0.245, *p* < 0.001), supporting H_6_. Overall, our predictors managed to explain a sizeable portion of the purchase intention for organic food (*R*^2^ = 0.321). Figure 3 and Table 4 summarize the results for the structural model.

## 5. Discussion

In our study, we set out to understand the purchase intention for organic food in a discount grocery setting. Our findings support most of the hypothesized effects in our proposed model. The effect of health consciousness on the purchase intention is by far the strongest and is generally in line with organic food purchasing behavior found in the existing literature [41,43,60]. Similarly, our findings are in line with the literature for environmental concern and its positive relationship with the purchase intention for organic food [35,36]; this seems to also hold true in this discount grocery shopping setting. The positive effect of health consciousness (β = 0.462) that we found in this discount setting is far stronger than the effect of environmental concern (β = 0.130) on the purchase intention for organic food, perhaps hinting at a stronger self-orientation (toward one’s own health) of discount shoppers rather than an altruistic focus (toward the environment). The hypothesized negative effect of price consciousness on the purchase intention was also supported. The effect—especially if viewed in conjunction with the positive relationship we found between hedonic shopping value and the purchase intention—is not overly surprising, as it hints at an underlying notion that organic food is indeed a premium product, and discount grocery shoppers act accordingly, which may also be reflected in the lack of a significant relationship between price consciousness and hedonic shopping value. Overall, the positive effect of hedonic shopping value on purchase intention supports other studies that demonstrate that utilitarian shoppers may move toward hedonic shopping behavior [5]. Additionally, the negative effect of price consciousness is in line with previous studies finding that price can act as a barrier in organic purchase situations [32,61] and holds true in the discount grocery setting of our study. In the said discount grocery setting, this leads us to conclude that price consciousness seems to still be a highly important factor in purchase decisions for shoppers—even for premium products such as organic food [47]. But when comparing the magnitude of the regression weights in our structural model, we find that health consciousness and hedonic shopping value have a far stronger impact on the purchase intention for organic food in this discount setting. This leads us to conclude that while it is an important factor, it is not all about the price.

### 5.1. Managerial Implications

The findings of this study hold several implications for discount grocery retailers, which lead to three courses of action. (1) As the relationship between price consciousness and purchase intention was found to be negative, discount grocery retailers might consider looking into the comparative case, that is, further cultivating the image that even premium products, such as organic food, can be bought at a lower price at their supermarkets. Additionally, however, as environmental concern and health consciousness are significant drivers of the purchase intention for organic food, discount grocery retailers might also benefit from specifically promoting these factors, even in this discount setting. (2) Our findings suggest that the greener image that discount grocery retailers have been seeking to portray may be positively received by their customers, as we found a positive relationship between environmental concern and the purchase intention for organic food in the discount setting. The implication for discount grocery retailers can be that these image investments are paying off in a low-price environment and may thus be worth maintaining. (3) And finally, our findings support the notion that addressing the hedonic side of consumers does indeed have merit in the discount setting. Nevertheless, traditional players like Aldi and Lidl should be careful to strike the right balance between a more upscale image and the traditional no-frills approach, especially as other competitors enter this new void in the German home markets by strictly focusing on the traditional no-frills (i.e., hard discount) approach, from which Aldi and Lidl are moving away.

### 5.2. Contribution and Future Research Areas

This paper contributes to the existing literature in a number of ways. This study looks at the purchase intention for organic food in a discount setting, linking it to price consciousness. Additionally, in this discount grocery setting, it tests the established drivers of organic purchase intention, environmental concern, and health consciousness. Furthermore, this study adds to the literature [5] by looking at hedonic value in a discount setting. Still, as a result of our findings, we would envision six potential avenues for future research: (1) We would urge researchers to conduct willingness to pay (WTP) studies in a discount grocery context to understand the organic premium consumers are willing to pay in a discount setting. (2) Similarly, and perhaps more broadly, we could envision contingent valuation studies attempting to analyze individual organic food attributes and their corresponding prices, especially in a comparative setting between discount and non-discount (i.e., full-range) grocery retailers. (3) Future studies could attempt to further understand price consciousness in the discount setting by studying its effect on different products and product categories. (4) Future research might also attempt to examine the aspect of hedonic shopping value in light of new entrants in the discount retailing market, who cultivate a more traditional, hard discount offering. (5) Additionally, we could envision other researchers expanding our model by introducing other factors related to the purchase intention for organic food. (6) Lastly, future studies could attempt to replicate the findings of this study using different countries, an experimental design in-store, or actual purchase data.

### 5.3. Limitations

In conclusion, some limitations of this study should be mentioned. Firstly, the choice of constructs may be a limitation. While environmental concern and health consciousness are often employed in organic food research [16], price consciousness is traditionally associated with discount grocery retailing [6], and while hedonic shopping value has played a role in the recent discount shift [5], additional and/or other constructs may be of interest in studying organic food purchases in a discount grocery setting. This, however, may be a topic for future research, as outlined in the previous section. Secondly, it should be noted that discount grocery shoppers are not necessarily distinctly different from other grocery shoppers: they may merely exhibit certain traits in a more or less pronounced manner. Lastly, for this study, we recruited US respondents, and we, therefore, acknowledge the possibility that our findings may not hold true for individuals with different cultural backgrounds for multiple reasons, such as scaling biases in the survey items.

## 6. Conclusions

In our study, we set out to shed light on the purchase intention for organic food in a discount grocery retail setting. In a structural equation model based on a US consumer survey (*n* = 394), we found a positive relationship between hedonic shopping value, environmental concern, as well as health consciousness and the purchase intention for organic food. Furthermore, our results show that price consciousness exhibits a direct and negative relationship with purchase intention. Additionally, we found that that the impact of environmental concern, health consciousness, and hedonic shopping value is greater on the purchase intention of organic food than that of price consciousness—even in this discount setting.

## Figures and Tables

**Figure 1 foods-09-00458-f001:**
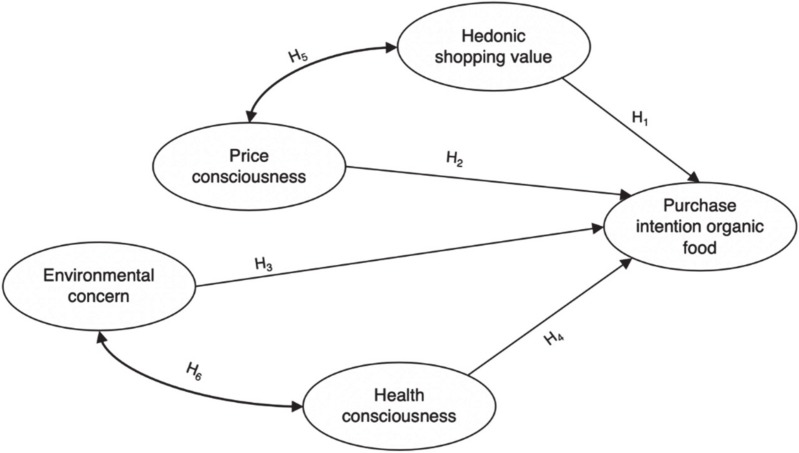
Conceptual framework. Note: H_1_ to H_6_ denote the stated hypotheses.

**Figure 2 foods-09-00458-f002:**
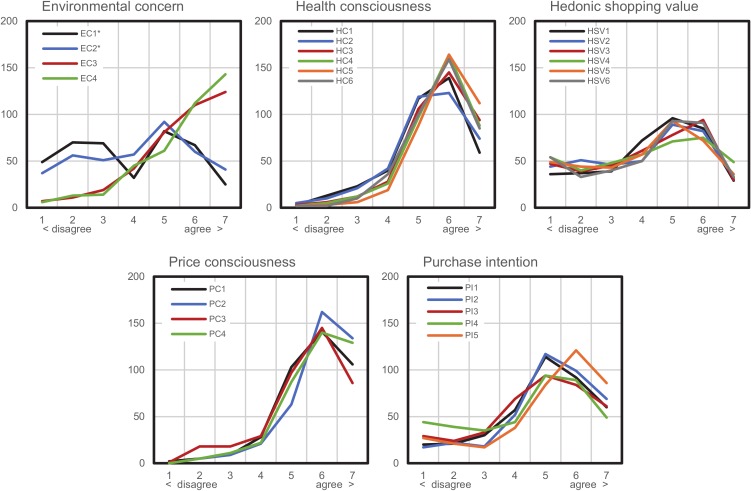
Overview of responses (*n* = 394). Notes: *y*-axis: number of responses; *x*-axis: 7-point scale (1 = strongly disagree, 2 = disagree, 3 = somewhat disagree, 4 = neither agree nor disagree, 5 = somewhat agree, 6 = agree, 7 = strongly agree); * reverse-coded items.

**Figure 3 foods-09-00458-f003:**
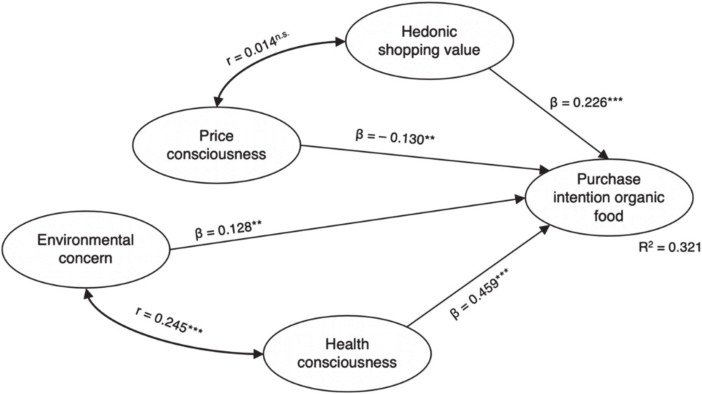
Results for structural model. Notes: β = standardized regression coefficient; *R*^2^ = coefficient of determination; *** *p* < 0.01; ** *p* < 0.05; n.s. = not significant.

**Table 1 foods-09-00458-t001:** Sample descriptive statistics.

Demographics	Sample (*n* = 394)	
Gender		
Male	221	56%
Female	173	44%
Age range		
≤25	55	14%
26–30	91	23%
31–35	87	22%
36–40	70	18%
41–45	30	8%
46–50	28	7%
>50	33	8%
Mean	35.4	
Education		
High school diploma	77	20%
Vocational training	21	5%
Bachelor’s degree	234	59%
Master’s degree or PhD	57	14%
Other	5	1%
Annual household income		
up to USD 20,000	39	10%
USD 20,001–40,000	87	22%
USD 40,001–60,000	108	27%
USD 60,001–80,000	80	20%
USD80,001–100,000	50	12%
over USD 100,000	40	10%

**Table 2 foods-09-00458-t002:** Measurement of constructs.

Item	Factor Loading
Environmental concern (EC) (Cronbach’s α = 0.736), *adapted from Wei, Ang, and Jancenelle* [49] *as well as De Magistris and Gracia* [48]	
EC1: Environmental problems are not affecting my life, personally. *	0.766
EC2: I can think of many things I’d rather do than work toward improving the environment. *	0.702
EC3: The current development path is destroying the environment.	0.739
EC4: Unless we do something, environmental damage will be irreversible.	0.797
Health consciousness (HC) (Cronbach’s α = 0.902), *adapted from Michaelidou and Hassan* [54]	
HC1: I reflect about my health a lot.	0.760
HC2: I’m very self-conscious about my health.	0.794
HC3: I’m alert to changes in my health.	0.881
HC4: I’m usually aware of my health.	0.797
HC5: I take responsibility for the state of my health.	0.731
HC6: I’m aware of the state of my health as I go through the day.	0.869
Hedonic shopping value (HSV) (Cronbach’s α = 0.955), *adapted from Babin et al.* [26]	
HSV1: A shopping trip is truly a joy.	0.908
HSV2: I usually continue to shop not because I have to, but because I want to.	0.856
HSV3: Compared to other things I could do, the time I spend shopping is truly enjoyable.	0.945
HSV4: I enjoy shopping trips for their own sake, not just for the items I may purchase.	0.901
HSV5: During shopping trips, I feel the excitement of the hunt.	0.908
HSV6: While shopping, I feel a sense of adventure.	0.906
Price consciousness (PC) (Cronbach’s α = 0.801), *adapted from Gil and Soler* [53]	
PC1: I try to buy food items that are on sale.	0.845
PC2: I pay attention to good deals.	0.793
PC3: I remember the prices I’ve paid before.	0.638
PC4: I compare food prices from different brands.	0.797
Purchase intention for organic food (PI) (Cronbach’s α = 0.935), *adapted from Yadav and Pathak* [55]	
PI1: I will purchase organic food for personal use.	0.889
PI2: I am willing to purchase organic food for personal use.	0.859
PI3: I will make an effort to purchase organic food.	0.886
PI4: I have been purchasing organic food on a regular basis.	0.902
PI5: I have purchased organic food over the past six months.	0.920

* reverse-coded.

**Table 3 foods-09-00458-t003:** Overview of reliability and validity measures.

	CR	AVE	MSV	HSV	HC	PI	PC	EC
**HSV**	0.956	0.785	0.120	0.886				
**HC**	0.905	0.613	0.249	0.263	0.783			
**PI**	0.937	0.748	0.249	0.346	0.499	0.865		
**PC**	0.811	0.521	0.155	0.014	0.394	0.093	0.722	
**EC**	0.772	0.555	0.092	0.008	0.245	0.203	0.303	0.745

**Table 4 foods-09-00458-t004:** Results for the structural model.

Hypothesis and Path				Std. Regression Weights/Correlation		*p* Values	Supported
H_1_	Hedonic shopping value	⟶	Purchase intention organic food	β =	0.226	*p* < 0.001	yes
H_2_	Price consciousness	⟶	Purchase intention organic food	β =	−0.130	*p* = 0.022	yes
H_3_	Environmental concern	⟶	Purchase intention organic food	β =	0.128	*p* = 0.016	yes
H_4_	Health consciousness	⟶	Purchase intention organic food	β =	0.459	*p* < 0.001	yes
H_5_	Hedonic shopping value	⟷	Price consciousness	*r* =	0.014	*p* = 0.809	no
H_6_	Environmental concern	⟷	Health consciousness	*r* =	0.245	*p* < 0.001	yes

Model fit: χ^2^ = 712.034, df = 242, TLI = 0.924, CFI = 0.933, RMSEA = 0.070, SRMR = 0.054. Explanatory power of model: purchase intention for organic food: *R*^2^ = 0.321.

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
