# Peer review of "Is It All about the Price? An Analysis of the Purchase Intention for Organic Food in a Discount Setting by Means of Structural Equation Modeling"

_foods, 2020, doi:10.3390/foods9040458_

Round 1

Reviewer 1 Report

This is a well-written paper with a working model. However, I have a few concerns regarding the goals of the paper, the hypotheses and the methodology.

Major suggestions

My first concern is regarding the hypotheses. Hypotheses like “Price consciousness has a direct and negative effect on the purchase intention for organic food.”, „Environmental concern has a direct and positive effect on the purchase intention for organic food.”, „Health consciousness has a direct and positive effect on the purchase intention for organic food.” are too obvious. Let me give you some solutions as I suppose the real hypotheses were not as – sorry to say – boring as these hypotheses. E.g. these hypotheses can contain that ‘even in discount grocery setting’ this is the situation. Or you can hypothesize that this is the situation in the US as well. Let me also suggest you express as a goal of the research to compare the effects of these effects on purchase intent. I do not agree with hypothesizing something which is already written in the literature. You write e.g. that “(…) environmental challenges caused by human behavior, is an oft-cited driver for organic food purchase behavior with a positive influence on purchase intention”. So, if we already know that this is true as it is an “oft-cited driver” why do you hypothesize this? Also as a second example, you state that “Several studies have not only found health consciousness to be a driver for organic food purchases, but (…)” and after that, you hypothesize that “Health consciousness has a direct and positive effect on the purchase intention for organic food”. If we already know, we should not hypothesize it.

The Methodology is not enough detailed. It seems to be important for the goals of the paper to ask only respondents who do anytime their grocery shopping in discounts, but as the sampling method is not clearly introduced the reader can not understand whether respondents were main grocery shoppers or not, whether respondents do their grocery shopping anytime in discounts or not, whether they buy organic foods anytime or not.

It is interesting why authors added mean value for “annual household income” to Table 1. I do not think this helps anyhow to better understand the sample and to rely on the dataset. I recommend authors to add income level groups or simply delete this part from the table.

At the beginning of the Results chapter authors write the following “Figure 2 visualizes the responses received, showing a homogenous response pattern for all constructs except for environmental concern, which may be explained by employing two reverse-coded items.” It is not clear to me why authors think that because of using reverse-coded items results can be different. This needs some explanations.

The model is nice and looks to be well-working.

Authors state in the “Managerial implications” chapter, that “This study holds several implications not only for discount grocery retailers, but for all grocery retailers.” I think that – if the sample was from discount grocery shoppers as I suppose – authors should not give suggestions for all grocery retailers.

After “Limitations” chapter authors should add “Acknowledgement”, “Author contributions” and “Conflicts of Interest” chapters as well.

Minor suggestion

In lines 86, 146, 162, 180 “Error! Reference source not found” appear.

Author Response

Dear reviewer 1,

Thank you very much for your time and for your remarks. Please refer to attached pdf-file for our reply.

Kind regards

Felix Katt and Oliver Meixner

Reviewer 2 Report

The article “Is it all about the price? An analysis of the purchase intention for organic food in a discount setting by means of Structural Equation Modeling” deals with a new topic, the purchase of organic food in discount grocery retailers. The article is rigorous in structure, content and methodology, but I would like to make some considerations that I would appreciate if the authors took it into account:

Regarding the conceptual model:

1.-  Hypothesis H1 is not sufficiently justified. Furthermore, could the authors explain in more detail the content of the hedonic value of the purchase, specifically, applied to the purchase in discount grocery retailers? That is, what are the characteristics of the hedonic value of purchase in discount grocery retailers?

2.- In my opinion, the interactions are little justified. Why these correlations? What literature supports these correlations? Why can't other correlations be?

Regarding the methodology

1.- The authors should explain a little more how to collect the information, Amazon Mechanical Turk (MTurk), since it is little known, mentioning its reliability compared to other methods.

2.- Regarding the age of the respondents, why are there fewer buyers from the age of 41? see Table 1

It is possible that by having a sample of young people, the H1 and H2 are corroborated since the young people have a greater health awareness and concern for the environment,

As for the references,

1.- I was surprised to find this message: "Error. Reference source not found" on lines 86, 146, 163 and 180. Could you explain it?

2.- Yadan and Pathak (2017) on line 179 should have been replaced by [54]

Author Response

Dear reviewer 2,

Thank you very much for your time and for your remarks. Please refer to attached pdf-file for our reply.

Kind regards

Felix Katt and Oliver Meixner

Round 2

Reviewer 1 Report

Dear Authors,

thank you for taking my recommendations strictly.

Author Response

Dear Reviewer 1,

thank you very much for your efforts.

Kind regards

Felix Katt & Oliver Meixner

Reviewer 2 Report

I acknowledge and appreciate the effort and speed of the authors to correct the deficiencies of their article, but I regret having to insist on the following aspects:

1.- The justification of hypotheses H5 and H6 is insufficient. The H5 has only been justified with this sentence "Given that discount grocery retailers main customer base was and probably still is price-consciousconsumers [6]" and one reference. Are there  no other reasons? Is there no more literature? Is this a new hypothesis? If so, perhaps it should be mentioned

With regard to the H6, it is not even justified, it is only stated and one reference is added, which is also an article that makes a review. Perhaps the authors could read Shamsolla and Juvancic (2010), and Shamsolla et al (2013), Tsakiridou et al. (2008) or search for other specific literature,

2.- The authors write "to ensure reliable and valid results, we followed previous researchers' guidelines for ..." (lines 173-174) who are these researchers? The justification for the method of collecting information remains insufficient.

3.- Perhaps this information "mean age of US population = 38.2" should be included in the article so that the reader does not have the same doubt that I have had.

I again thank the authors for their effort, and I greet them sincerely.

Author Response

Dear reviewer 2,

Thank you very much for your time and for your remarks. We have attempted to address your comments in full and will briefly attempt to summarize the changes we have undertaken:

  • Regarding the justification of hypotheses H5 and H6, we have added additional sources, Carpenter and Moore (2009), Tsakiridou et al. (2008), De Magistris and Gracia (2008), as well as Gschwandtner (2018) to underpin these proposed secondary hypotheses [lines 149-156].
  • Regarding the guidelines for MTurk, we have made the sources of these frequently used MTurk guidelines, Mason and Suri (2011) as well as Peer et al. (2013), more visible. Additionally, we have pointed out that MTurk is commonly used as a means to recruit respondents in organic food surveys [lines 172-174].
  • Regarding the mean age of the US population, we have added this information to the text [line 185].

We hope that we have correctly interpreted your remarks and would like to once again thank you for your time and consideration!

Kind regards

Felix Katt & Oliver Meixner